# Phototunable self-oscillating system driven by a self-winding fiber actuator

Zhiming Hu [1,2], Yunlong Li[1,2] & Jiu-an Lv [1,2✉]

Self-oscillating systems that enable autonomous, continuous motions driven by an unchanging, constant stimulus would have significant applications in intelligent machines, advanced robotics, and biomedical devices. Despite efforts to gain self-oscillations have been made through artificial systems using responsive soft materials of gels or liquid crystal polymers, these systems are plagued with problems that restrict their practical applicability: few available oscillation modes due to limited degrees of freedom, inability to control the evolution between different modes, and failure under loading. Here we create a phototunable self-oscillating system that possesses a broad range of oscillation modes, controllable evolution between diverse modes, and loading capability. This self-oscillating system is driven by a photoactive self-winding fiber actuator designed and prepared through a twistless strategy inspired by the helix formation of plant-tendrils, which endows the system with high degrees of freedom. It enables not only controllable generation of three basic self-oscillations but also production of diverse complex oscillatory motions. Moreover, it can work continuously over 1270000 cycles without obvious fatigue, exhibiting high robustness. We envision that this system with controllable self-oscillations, loading capability, and mechanical robustness will be useful in autonomous, self-sustained machines and devices with the core feature of photomechanical transduction.

[1] Key Laboratory of 3D Micro/Nano Fabrication and Characterization of Zhejiang Province, School of Engineering, Westlake University, Hangzhou, Zhejiang Province, China. [2] Institute of Advanced Technology, Westlake Institute for Advanced Study, Hangzhou, Zhejiang Province, China. ✉email: lvjiuan@westlake.edu.cn

The intelligent and autonomous features of living organisms have inspired truly innovative achievements in devising smart materials that allow a variety of motions in response to an external stimuli[1,2]. Autonomous self-oscillations are ubiquitous in living organisms and enable the working of various important life activities, such as animal's heart beating, birds' wingbeats, and cell cycling[3–5]. All these biological oscillations rely on non-equilibrium dynamics, which are sustained by the constant input of chemical energy, and do not need any switching on/off stimuli[6]. However, existing man-made stimuli-responsive systems have mainly focused on the generation of equilibrium or metastable structures/states, in which the switching of structures/states always requires the on/off trigger, and autonomous, self-sustainable behavior is largely absent[5]. As a result, only unsustainable single motion can be gained under constant stimulation in these systems[3]. Scientists and engineers have attempted to realize autonomous self-oscillations in synthetic systems mostly using two kinds of stimuli-responsive soft materials. One is smart gels that is driven by pH or Belousov–Zhabotinsky reaction[7] and the other is photoactive liquid crystal polymers (PLCPs)[6,8–11]. However, the former must be operated in wet conditions, because their driving force is induced by swelling and deswelling of the stimuli-responsive gels, whereas working in a dry environment is preferred in most practical applications[4,6]. For the latter, PLCPs are often processed into freestanding strips as oscillators, which employ the self-shielding effect to produce a feedback loop of bending/unbending to gain oscillation[8], and the bending is largely demonstrated as the main degree of freedom (DOF). It is challenging to achieve diverse oscillating modes in a single PLCP actuator, because more DOFs are required. In particular, it turns more challenging when controllability of various oscillations in a single actuator is targeted to achieve[9]. In addition, most previous oscillating systems fail to work under loading, because the loading changes and affects their equilibrium conditions, and hinders self-oscillations, whereas real engineering applications need to work against an external load[12]. Therefore, intelligent stimuli-responsive materials with high DOF and loading ability are highly needed to develop versatile self-oscillating systems with controllable oscillations and working capability towards practical applications.

Shape-change materials exhibiting deformation behavior with high DOF are widespread in nature, particularly in plants[13,14]. Shape changes from a straight structure to a helically coiled structure in plants display diverse deformation behaviors with high DOF, such as bending, twisting, coiling, and winding. The formation of helix structure in plant tendrils is one of such shape changes (Fig. 1a), which has fascinated scientists for centuries[15,16], because it allows morphing of simple, one-dimensional materials into complex three-dimensional (3D) helical geometries. The mechanism of the helix formation can be illuminated by a simplified physical model based on a filament composed of two layers with a differential growth rate[17]. As shown in Fig. 1b, the differential growth, arising from faster growth of the top layer and lower growth of the bottom layer, induces intrinsic curvature that leads to spontaneous coiling and winding, and results in the formation of helices to avoid steric interactions when $L_0 > 2\pi R$ (Fig. 1c, $L_0$ is the length of the filament and $R$ is the radius of the arc of the curved filament)[18]. The shape change in the helix formation appears as a result of the intrinsic curvature, which develops owing to an asymmetric distribution of strain/stress over the cross-sections of the filament[18,19].

Inspired by this biological mechanism, herein we designed a self-winding fiber actuator (SWFA) for a phototunable self-oscillating system (PSOS). The premise of our design is that the asymmetric distribution of strain/stress over cross-section must be built and embedded into its own material architecture of SWFA, which enables it to self-shape, morph, and actuate by following the shape change from a straight structure to a helically coiled structure. The design was based on three criteria as follows: (1) SWFA must achieve the reversible shape change between a straight structure into helically coiled structures, offering deformation behaviors with high DOF; (2) SWFA must quickly dissipate the absorbed light through heat by deformation, which would help to generate out-of-equilibrium dynamics[8]; and (3) SWFA must work under loading.

## Results and discussion

The first requirement is satisfied by using the plant-tendril-inspired strategy described above. Widely used strategies for the introduction of artificial helical coils in soft actuators is through twist insertion in highly stretched orientated polymer fiber[20]. Whereas plant tendrils, on the contrary, show a twistless mechanism[15] that relies on the intrinsic curvature rather than twist insertion (Fig. 1b, c). To gain the asymmetric distribution of strain/stress over the cross-section of a fiber actuator, we start the fabrication using a precursor of helically coiled soft spring (Fig. 1d, e) that possesses a differential in length between its outer and inner circumference (Fig. 1f). This precursor is prepared through employing a screw mold (Supplementary Fig. 1) to shape a liquid crystal elastomer (LCE) oligomer into the helically coiled soft spring with a well-defined weak network formed by chemical crosslink reaction (see details in the "Methods" section). After initial curing, the shaped soft spring, in which the chemical crosslink reaction is not completed and still in progress over time, is removed from the screw mold and used as the precursor. Next, the fresh prepared precursor is axially stretched (Fig. 1g). Upon stretching, the stress accumulated in the inner side is much greater than the outer side of the soft spring because of their difference in length. After the stretching operation, the stress gradient over the cross-sections of the straightened spring fiber is induced and fixed via the chemical crosslink reaction (see details in the "Methods" section). It is worth mentioning that, instead of directly mimicking the biological model based on the bilayer filament structure used by previous studies[17,21], our strategy employs a single phase, which offers a straightforward and scalable alternative without the need for precise design of the bilayer structure, and hence avoiding complicated material design and preparation[22]. As shown in Fig. 1h, the prepared SWFA exhibits diverse deformation behaviors (twisting, bending, coiling, winding, and tightening) upon near infrared (NIR) irradiation (Supplementary Movie 1). To satisfy the second requirement, a main-chain LCE doped with graphene that owns superior photothermal performance[23] is used as artificial photonic muscle material[24] to build SWFAs, to offer fast and powerful photo-mechanical actuation (Supplementary Fig. 7). The last requirement is satisfied via the integration of the external load as a part of an oscillating system, such as the classical block-spring oscillator. The external load connects and works together with a photoactive "soft spring" (SWFA) to generate oscillating motions.

From these principles, we have fabricated the PSOS, which is composed of three building elements as follows: the SWFA works as a photoactive "soft spring," a hanging object serves as the load, and an NIR light source provides the power and control (Fig. 2a). This system with structural simplicity and functional effectiveness not only exhibits three basic self-oscillations (Fig. 2b–g and Supplementary Movies 2, 3, and 4)—tilt oscillation, rotational oscillation, and up-and-down oscillation—but also enables locally and reversibly switching among these three self-oscillations (Fig. 2h and Supplementary Movie 5). The necessary condition to gain self-oscillations in PSOS is that the width of the light spot

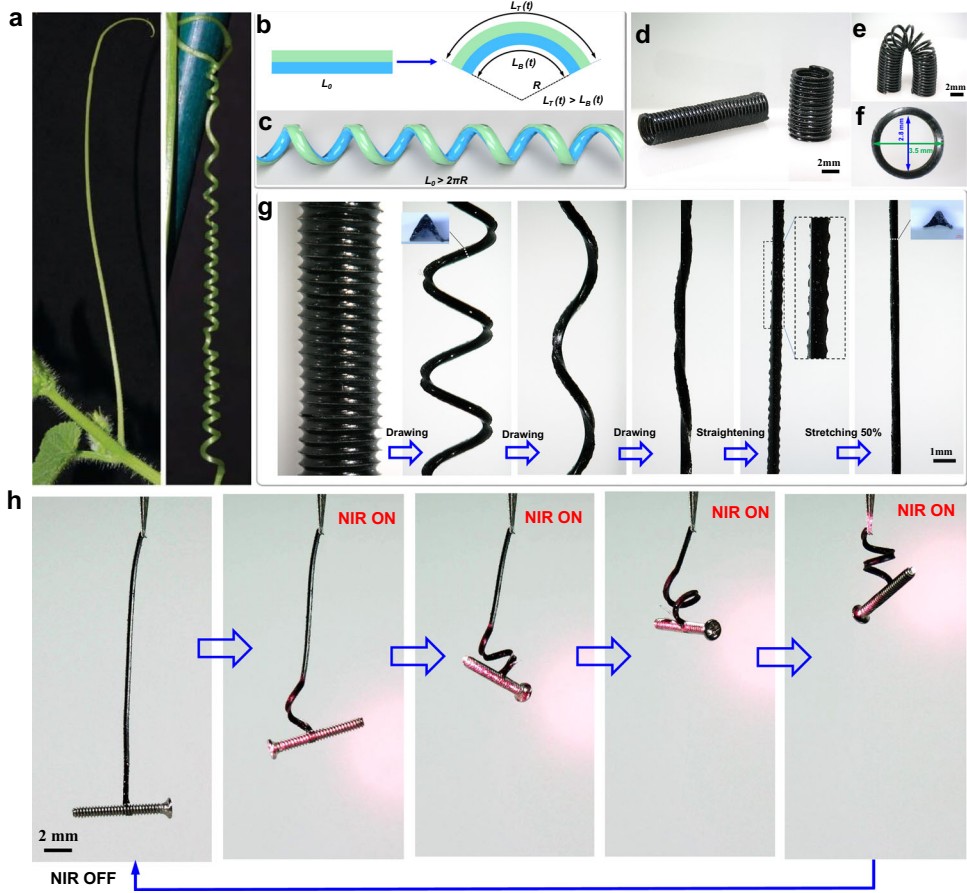

**Fig. 1 Bioinspired design of self-winding fiber actuator. a** Photographs showing a straight tendril (left) and a coiled tendril (right). **b, c** A simplified physical model describing how the helical coiling of plant tendril is formed. In this model, the tendril is simplified as a filament composed of two layers. Differential growth in the two layers induces intrinsic curvature that leads to spontaneous formation of helices (**c**) to avoid steric interactions when the length of the filament is long enough ($L_O > 2\pi R$). $L_O$ is the initial length and $R$ is the radius of the arc of the curved filament. $L_T(t)$ and $L_B(t)$ is the length of the top layer and the bottom layer of the curved filament after growing for a time $t$, respectively. **d, e** Photograph showing the precursor of helical soft springs used for the preparation of the self-winding fiber actuators. **f** Photograph showing a coil of the helical soft spring from the top view. The outer and inner diameter of the coil is 3.5 mm and 2.8 mm, respectively. **g** Sequential photographs showing the shape changing of a stretched soft spring during the fabrication of a self-winding fiber actuator. When the soft spring is stretched, it unwinds to turn a straight fiber, it can be observed that one lateral side of straightened fiber is smooth, while the other lateral side has wrinkles owing to their difference in length. Following 50% stretching smooths the wrinkled side and keeping the stretching at room temperature for 24 h, then the self-winding fiber actuator (SWFA) is obtained. **h** Snapshots showing the shape change of SWFA from a straight structure to a helically coiled structure upon NIR irradiation (Supplementary Movie 1). The weight of the hanging object is 35 mg. The intensity of NIR light is ~4 W cm$^{-2}$.

irradiated on an SWFA must be smaller than the diameter of the coils induced by the light, while light intensity must also be above a threshold (Supplementary Fig. 8). In other words, the light-induced coils of the SWFA must be partially out of the light spot. If the width of the light beam is large enough, the formed coils will be fully within the light spot and the irradiated SWFA will be finally balanced into a stable coiled shape without oscillation.

To figure out the mechanism of the formation of the three basic oscillations, we observed photo-induced deformation of SWFA upon NIR irradiation. As shown in Fig. 3a, SWFA undergoes three different stages of deformation behavior as follows: Stage 1, the SWFA is contracting and self-twisting; Stage 2, the twisted SWFA is bending and winding to form a loose coil spring; Stage 3, continuous NIR irradiation shortens the distance between the coils and tightens the loose spring. All these deformation behaviors arising from light-triggered shape memory of SWFA tends to return to its initial helix shape. Time-varying morphologies of the SWFA and deformation behaviors in these three stages determine the oscillating modes of PSOS. In Stage 1, the twisting of SWFA enables rotational oscillation. In Stage 2, the

bending and winding of SWFA allow the system to produce tilt oscillation and rotational oscillation, respectively. In Stage 3, the shortening motion of the coiled SWFA endows the system with up-and-down oscillation. It should be noted that each stage is dominated by one type of deformation; however, during the transition between two successive stages, different types of deformation behaviors occur simultaneously and induce complex oscillations that combines diverse self-oscillations.

The width of the NIR light beam plays important role in determining the modes of self-oscillation. In the case with constant light intensity above a threshold, a narrower spot tends to trigger tilt oscillation and rotational oscillation. When the spot is wide enough, up-and-down oscillation will occur (Supplementary Fig. 9). The reason is that the narrower spot is prone to make SWFA stay in Stage 1 or 2, whereas a wider spot is more likely to trigger the SWFA to directly transition into Stage 3. Thus, by adjusting the width of the spot irradiated on a SWFA, oscillating modes can be actively tuned. Furthermore, by changing the intensity of the NIR light, the amplitude of self-oscillations can be tuned significantly, whereas oscillating frequency has only a slight

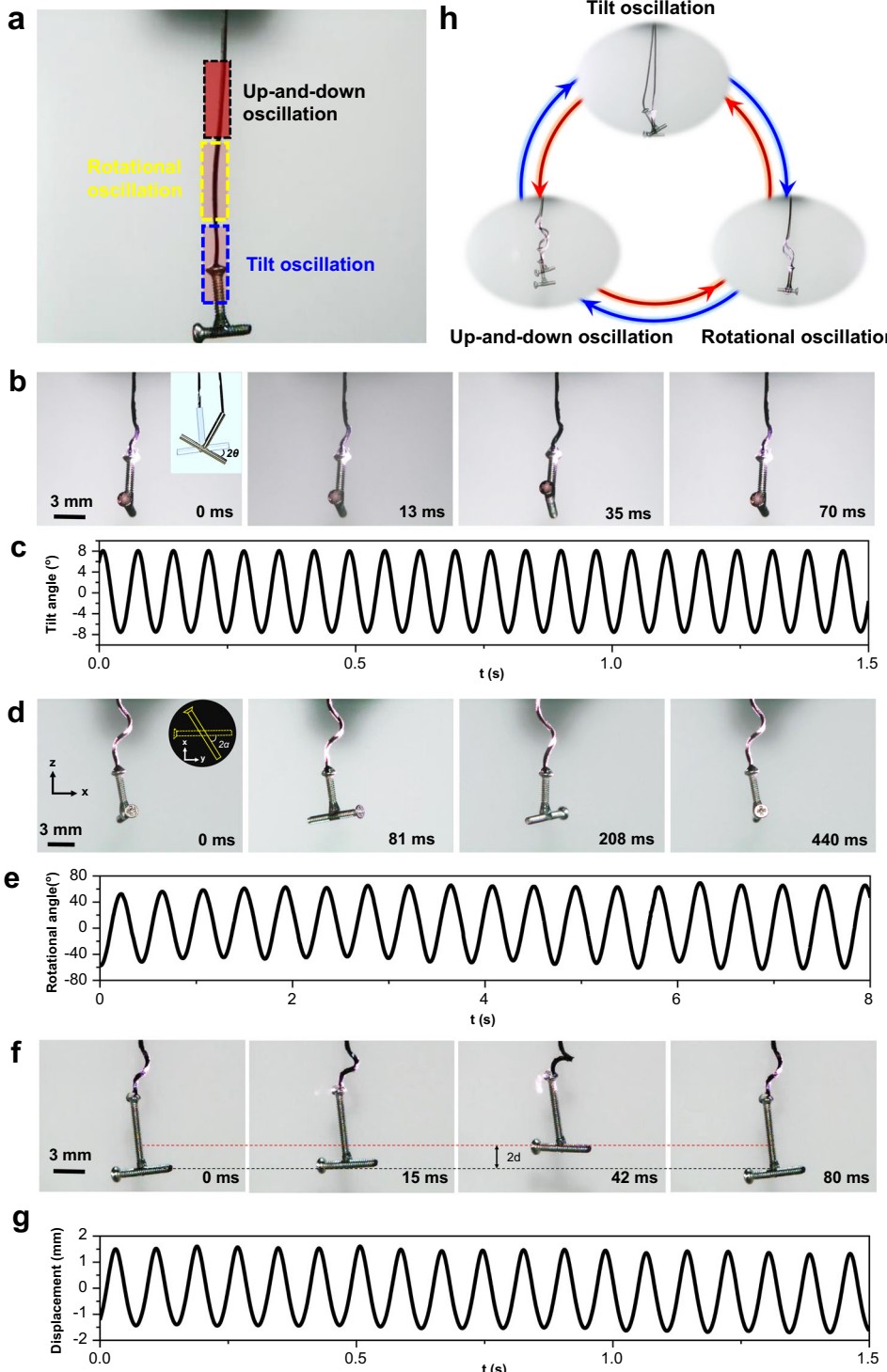

**Fig. 2 Phototunable self-oscillations. a** Photograph showing the location and intensity of NIR spot that triggers three basic self-oscillations. The shade of red color enclosed by dashed boxes schematically indicates the intensity of the light. With the same size of the NIR spot, the thresholds of light intensity must be met to generate three self-oscillations (see Supplementary Fig. 6). The threshold required to produce up-and-down oscillation is the highest among the three basic self-oscillations. To gain tilt oscillation, the connection part between the hanging object and the SWFA must be irradiated. **b, d, f** Snapshots experimentally showing three basic self-oscillations: tilt oscillation (Supplementary Movie 2), rotational oscillation (Supplementary Movie 3), and up-and-down oscillation (Supplementary Movie 4). **c, e, g** Tilt oscillation, rotational oscillation, and up-and-down oscillation are quantified through tilt angle ($\theta$), rotational angle ($\alpha$), and displacement ($d$) with time, respectively. **h** Photographs showing controllable and reversible switching among the three basic oscillation modes (Supplementary Movie 5). The T-shaped load is used to clearly exhibit self-oscillations. The weight of the hanging object is 62 mg. The intensity of NIR light is ~3.5 W cm$^{-2}$ for tilt oscillation and rotational oscillation, and ~5 W cm$^{-2}$ for up-and-down oscillation.

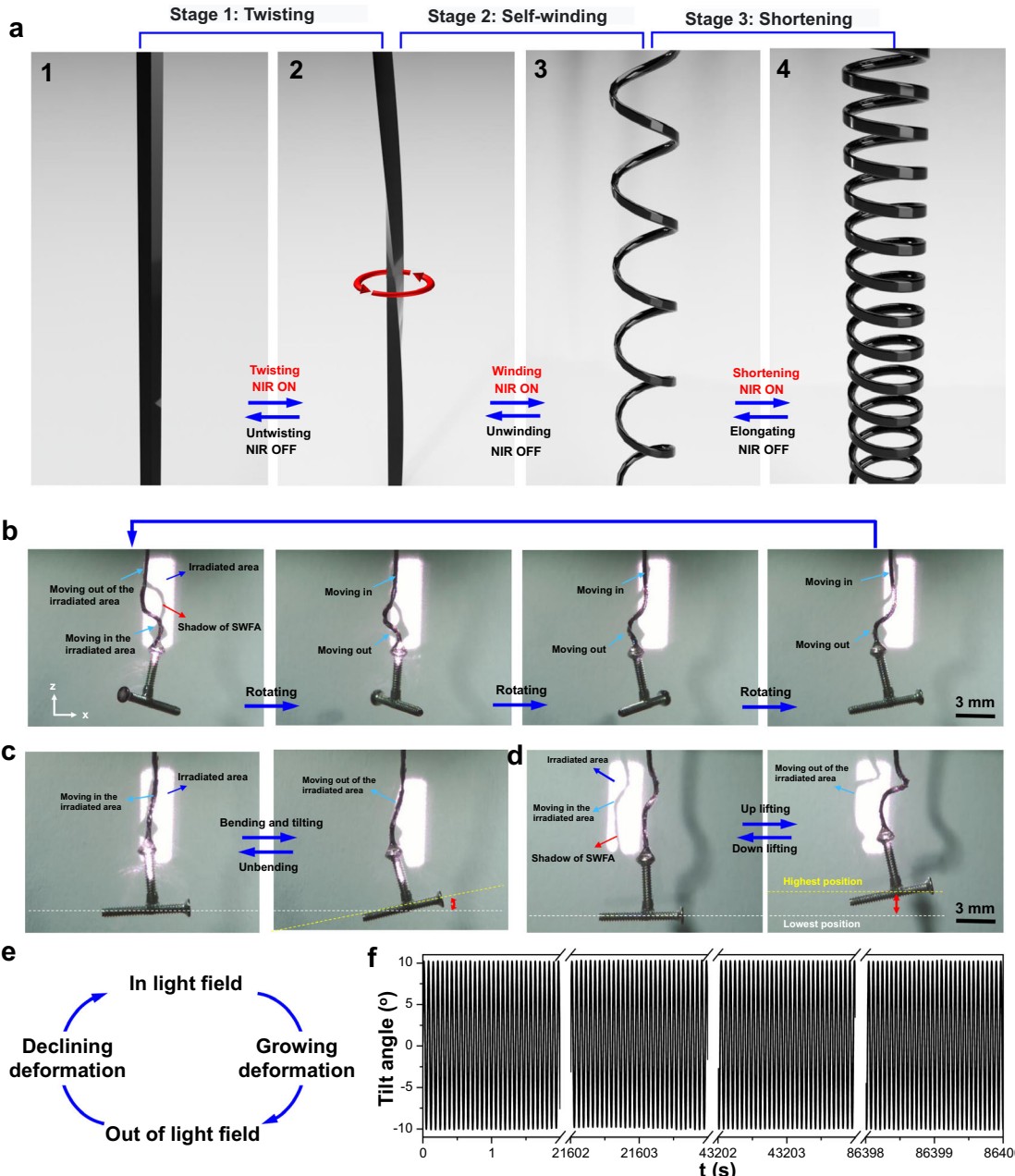

**Fig. 3 Mechanism of phototunable self-oscillations. a** Schematics showing the deformation and morphology change of SWFA in three stages upon NIR irradiation. It is assumed that the upper end of SWFA is fixed, while the lower end hangs a load. Stage 1, SWFA contracts and twists; Stage 2, the twisted SWFA bends and self-winds into coils; Stage 3, the coiled SWFA shortens the distance between the coils. **b, c, d** Snapshots recorded by high-speed camera showing the irradiated SWFA moving in and out of the light field during rotational oscillation, tilted oscillation, and up-and-down oscillation, respectively (Supplementary Movie 7). Through the location of the shadow of the irradiated SWFA in the light spot, the cycle motion of moving in and out of the light field can be clearly observed. **e** Feedback loop created by the cycle motion of moving in and out of the light field. **f** A total of 1,270,000 cycles of tilt oscillation in 24 h without obvious fatigue.

change (Supplementary Fig. 8), as the oscillating frequency of PSOS mainly depends on its intrinsic mechanical properties (the mass of loading, the modulus, and the size of the SWFA) rather than the input power of the NIR light.

To uncover the mechanism of the autonomous self-oscillations, infrared camera and high-speed camera were used to record the temperature change and the time-varying morphologies of SWFA, respectively. The NIR irradiation locally increases the temperature of SWFA (Supplementary Fig. 10 and Movie 6), owing to the photothermal excitation of the dopants. The internal heat triggers the deformation at the light spot. As shown in Fig. 3b–d, the deformation and morphological change of the SWFAs upon NIR irradiation make a part of the irradiated SWFAs moving out of the NIR field (Supplementary Movie 7). The partial SWFA keeps moving, while it is already out of the NIR field; this delay is attributed to the photothermal–mechanical transfer requiring time $\Delta t$ and the inertia gained during the actuation[8,9,25]. As moving away, the partial SWFA out of the light field cool down, leading it back to the light field. Then, the backed SWFA will repeat reversible motion in and out of the light field, forming a photomechanical feedback loop (Fig. 3e). The SWFA is self-propelled by repeating this feedback

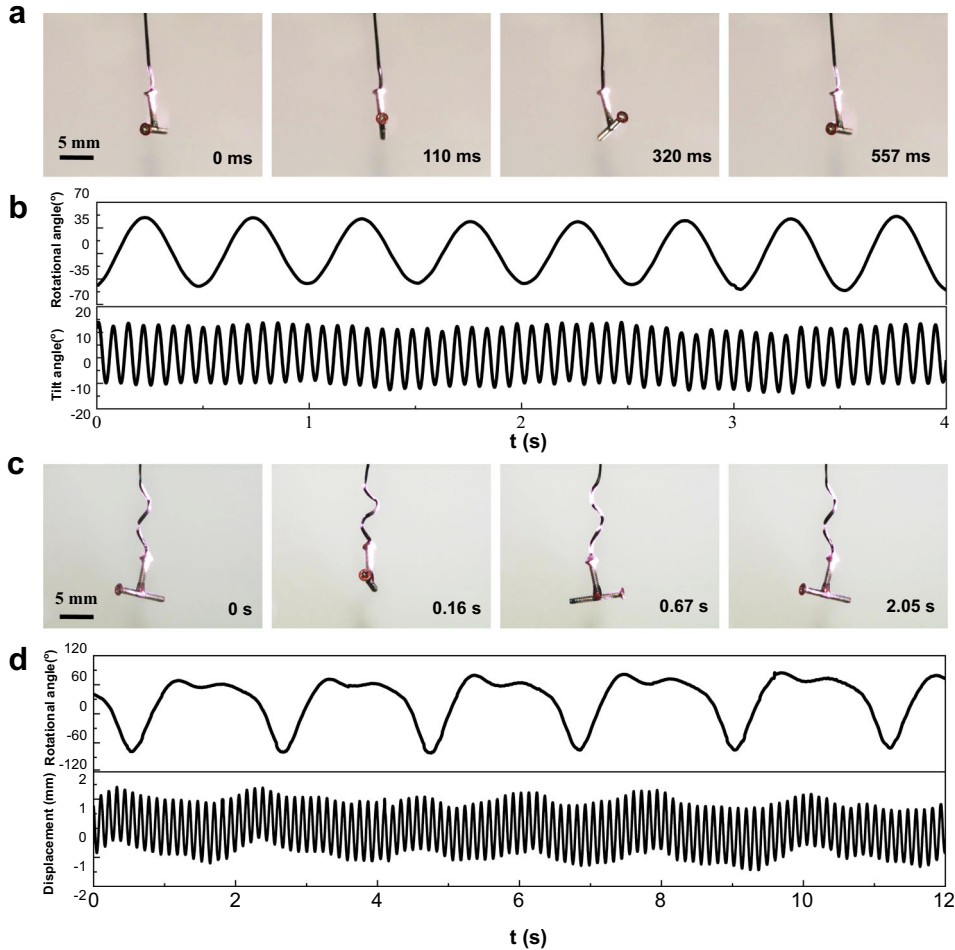

**Fig. 4 Complex self-oscillations. a, b** Snapshots exhibiting a complex oscillation that combines rotational oscillation and tilt oscillation (Supplementary Movie 9), and the corresponding oscillation dynamics. To gain this complex oscillation, the connection part between the hanging object and the SWFA must be irradiated. **c, d** Snapshots exhibiting a complex oscillation that combines rotational oscillation and up-and-down oscillation (Supplementary Movie 10), and corresponding oscillation dynamics. The intensity of NIR light is ~4.5 W cm$^{-2}$.

loop, which drives the PSOS to produce autonomous and sustainable oscillating motions. In our system, self-oscillations can be driven for a long time without obvious fatigue. For an example, as shown in Fig. 3f, the frequency and amplitude of tilt oscillation decrease from ~14.9 Hz to ~14.7 Hz and from ~10.2° to ~10.1°, respectively, after 24 h of continuous self-oscillation (1,270,000 cycles), exhibiting high reliability that is critically important for commercial applications.

In nature, living organisms take advantage of resonances to gain motion under natural/resonant frequency, in which they only consume minimum input energy but produce a large deflection amplitude, and achieve the highest energy efficiency[25]. PSOS exhibits a similar capability to amplify the amplitude of light-driven motion via resonance (Supplementary Fig. 11 and Movie 8). Moreover, living organisms can not only generate diverse self-oscillations, but also enable to synchronize multiple self-oscillations[26,27]. However, to our knowledge, artificial light-driven oscillating systems capable of transducing simple constant input into complex motion that synchronizes different oscillations have been seldom reported[28]. As shown in Fig. 4, PSOS enables generation and synchronization of different self-oscillations upon the constant irradiation of a single light beam (Supplementary Movies 9 and 10). In the face of multi-functional synchronization, these capabilities can effectively simplify the

control operation and real-time tunability by effectively reducing the number of control parameters[26].

Solar harvest and utilization are of great significance for modern industry and technological applications, as solar energy is a low-cost and inexhaustible power source[29]. We demonstrate that our self-oscillating system can capture and convert solar energy to generate autonomous periodic mechanical motion upon the irradiation of solar light without artificial participation (Supplementary Fig. 12a and Movie 11), which exhibits an effective and feasible approach to automatically and continuously extract energy from solar irradiation. Moreover, as a large part of the solar spectrum is composed of near-infrared light, we demonstrate that our system can convert NIR light to electrical energy via the principle based on Faraday's law of electromagnetic induction (Supplementary Fig. 12b, c). As shown in Supplementary Fig. 12d, the system allows continuous conversion of light to electricity lasting more than 2000 s without obvious fatigue, indicating its high reliability. This would be inspirable for scientists and engineers to design automated and self-driven systems for applications in remote power generation, solar energy harvest and transduce, and self-powered wireless sensors[30–33]. Furthermore, for real engineering applications, oscillating systems must conduct work in a damping media. PSOS can not only work in the air but also allow continuous and sustainable oscillations in

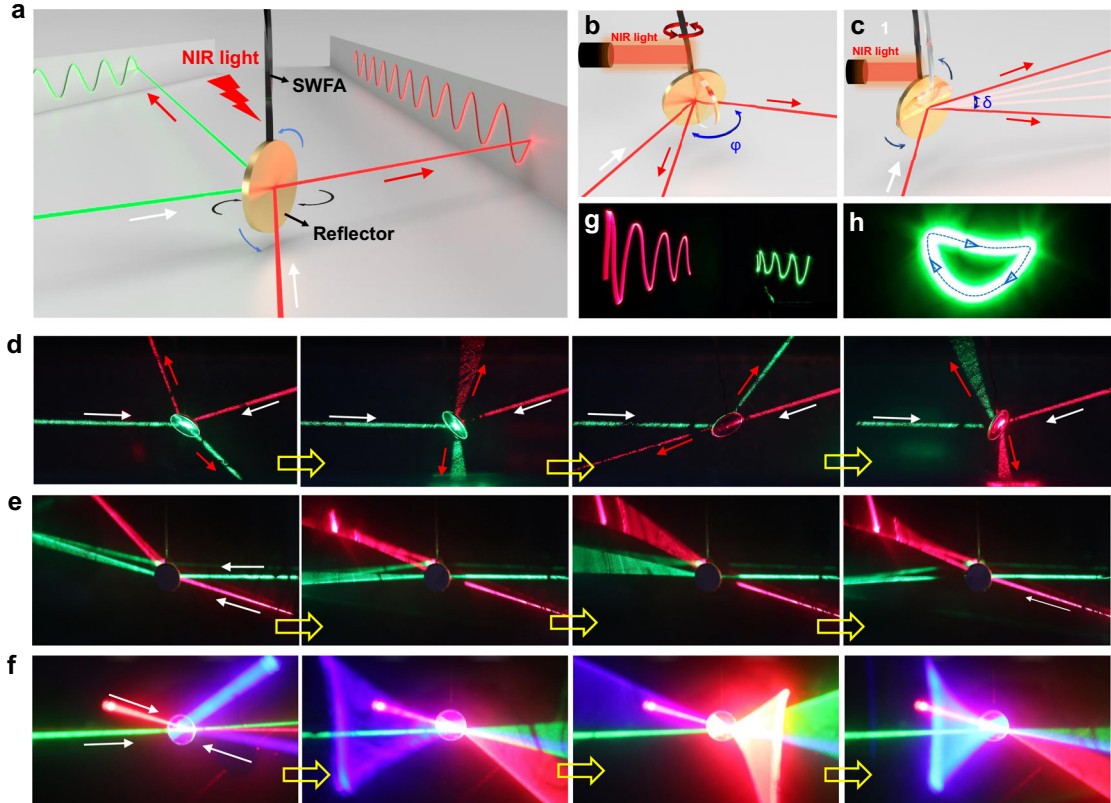

**Fig. 5 Laser modulation. a** Schematics showing the experimental setup of a laser steering system. Phototunable self-oscillations are used to control the periodic motion of the reflector, which is utilized to adjust the reflection direction of laser beams. **b**, **c** Schematics showing the horizontal scanning range ($\varphi$) and vertical scanning range ($\delta$), respectively. **d** Experimental photographs showing beam steering driven by rotational oscillation (Supplementary Movie 13). The reflection angle of the red and green laser beams was dynamically changed by the periodic rotation of the reflector. **e** Snapshots showing fast laser scanning driven by tilt oscillation (Supplementary Movie 14). Red and green laser beams were reflected and oscillated to generate fast scanning. **f** Snapshots showing 2D scanning of multiple laser beams using a complex oscillation (Supplementary Movie 15). The reflection direction of three laser beams (red, green, and blue laser beams) was dynamically changed by using the complex oscillation that combines rotational oscillation and tilt oscillation. **g** Sinusoidal trajectory of two moving laser spots. **h** Loop trajectory of a moving laser spot. The arrows indicate the moving direction of the laser spot. White and red arrows indicate the incident direction and reflection direction of laser beams, respectively.

high-damping mediums such as water (Supplementary Fig. 13 and Movie 12), which can effectively expand its application areas.

Moreover, we demonstrate that PSOS can be reconfigured to apply for modulation of laser beams. As shown in Fig. 5a, replacing the hanging object with a mirror reflector, we constructed a laser steering system that is composed of only three components: the mirror reflector functioning as the modulator, SWFA working as driving component, and a commercial NIR laser pointer serving as the power source and the controller. This system is a miniature, compact, lightweight, and of low power consumption, meeting the desired goal of optical systems required for beam modulation[34,35]. As shown in Fig. 5d–f, this steering system not only allows one-dimensional scanning with horizontal angular tuning range ($\varphi$) of ~360° (Supplementary Movie 13), substantially larger than that of ±50° achieved in recently developed LC beam steering devices[33], but also enables fast one-dimensional scanning with the vertical angular tuning range ($\delta$) of ~35° (Supplementary Movie 14), and even achieve two-dimensional scanning (Supplementary Movie 15). Moreover, as shown in Fig. 5g, h, the moving trajectory of laser spots can also be actively and locally adjusted via controllable self-oscillations. However, for traditional mechanical steering systems, the similar dynamic steering of the laser beams requires a large number of universal joints and rotating platforms in the mechanical steering system, which results in a bulky system with plenty of mechanical components and also, in turn, leads to

complex driving operation[36]. Our laser steering system featured with optical tunability, high efficiency, and wide tuning range could find use in applications that demand to dynamically control laser beams, such as 3D imaging and mapping, spatially resolved optical sensors, and freespace optical communications.

Phototunable oscillating systems reported in this report combines unique characteristics: controllable multimodal oscillations, working under loading, photomechanical robustness, and reconfigurable functions, which few other light-driven oscillating systems can achieve. The preparation method of SWFA, the key component for PSOS, is simple, scalable, and sustainable. Only a screw-shaped mold is needed to mold the soft spring precursor and then SWFAs can be produced simply through the bioinspired stretching treatment. The raw materials (liquid crystal monomers and photothermal dopants) for SWFA are abundant and commercially available chemicals. The entire manufacturing process of PSOS can be carried out in any materials lab without the need to use expensive facilities. In addition, the high robustness of autonomous self-oscillation of our system ensures long-term reliability, which is critically important when it would like to be integrated into consumer-based products. More importantly, by replacing the hanging object with different functional components, users can not only realize a variety of functions but also enable flexible reconfiguration of various functions in one system according to their purposes. It is anticipated that the development of this controllable, reconfigurable, and robust self-oscillating system would be of benefit in engineering and scientific applications, such as

sophisticate autonomous devices and systems, autonomous extraction of energy from solar irradiation, compact wireless scanners, and beyond.

## Methods

**General considerations.** RM82 (1,4-bis-[4-(6-acryloyloxyhexyloxy)benzoyloxy]-2-methylbenzene) was purchased from Shijiazhuang Yesheng Chemical Technology Co., Ltd. DODT (3,6-dioxa-1,8-octanedithiol), pentaerythritol tetrakis (3-mercaptopropionate) (PETMP), dipropylamine (DPA), and graphene were obtained from TCI. The surface temperature of the actuator in different oscillation modes before and after 808 nm NIR irradiation was measured by an infrared thermometer (A665sc, FLIR). NIR light (808 nm) was generated by a laser pointer (HW808AD1200-22FGD, Shenzhen Infrared Laser Technology Co., Ltd) or a laser source (PSU-H-LED, MDL-H-808-5W). The laser intensity was monitored by a laser power meter (TP100, Changchun New Industries Optoelectronics Technology Co., Ltd). The three lights used to demonstrate the steering function exhibited in Fig. 5 are 450 nm laser (blue), 532 nm laser (green), and 650 nm laser (red), which are produced by commercial laser pointers purchased from Beijing Huisite Technology Co., Ltd (green and red laser pointers) and Shenzhen Xinfei Optoelectronics Technology Co., Ltd (blue laser pointer). Supplementary videos were recorded by a super-resolution digital microscope (Keyence, VHX-1000C) or a digital camera (Canon, EOS 80D(W)). Slow motions of self-oscillations were recorded by a high-speed camera (Keyence VW-9000). Self-oscillating motions were tracked by kinematic analysis software (Kinovea).

**Preparation of the soft spring precursor.** The mixture was formulated with 1.67 : 1 molar ratio of RM82 and DODT, 3 : 1 molar ratio of DODT : PETMP, and 2 wt% graphene. Thiol groups and acrylate groups were equimolar. The mixture was dissolved in chloroform. After ultrasonic dispersion of 4 h, a catalytic amount of DPA was added into the solution. Then the reactant was dropped onto a screw mold (Supplementary Fig. 1) and quickly filled the space between threads via capillary force. After 2 h reaction at room temperature, the mixture on the screw-shaped mold was cured to form the soft spring, which was not fully chemically crosslinked. This soft spring will be used as the precursor to prepare the SWFA (Supplementary Fig. 2b, c).

**Preparation of SWFA.** Following the above preparation, the weakly crosslinked soft spring was drawn and untwisted to turn a straight fiber. Then the straightened fiber was stretched to gain 50% strain and it was maintained at 50% strain for 24 h to complete the curing through the chemical crosslink reaction between the acrylate monomer and thiol crosslinker. After the curing, the stretched fiber with uniaxial orientation was obtained and served as the self-winding fiber actuator.

## Data availability

All data supporting the findings of this study are available from the corresponding authors upon request. Source data are provided with this paper.

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

## Acknowledgements

This research was supported by Foundation of Westlake University, National Natural Science Foundation of China (51873197), and 151 Talent Project of Zhejiang Province.

## Author contributions

J.L. conceived the research. Z.H. and J.L. designed the experiments. Z.H. carried out the experiments. J.L. and Z.H. analyzed the experimental data. Y.L. did the tensile test. J.L. and Z.H. wrote the manuscript.

## Competing interests

The authors declare no competing interests.
