## [Peer Review File · Nature Communications]

REVIEWER COMMENTS

Reviewer #1 (Remarks to the Author):

I read the paper of Zhiming Hu on self-oscillating of spiral liquid crystal polymer fibers with great interest. The authors produced a spring-shaped actuator that is capable to provide continuous oscillation under a constant light source. The actuation figures could be adjusted as winding/unwinding and a continuous linear displacement. The authors based their system on by itself known liquid crystal oligomeric materials used by others for the fabrication of actuating systems (see the many liquid crystal elastomer (LCE) publications also mentioned by the authors. To make the actuators respond to light the authors blended the LCE oligomer with graphene to convert light absorption into heat, inducing order variations in the polymer. A special element in the system, which to my opinion deserves publication in Nat Comm, is the use of a screw-shaped mould to pre-structure the LCE fiber and the two-stage crosslinking basically freezing the LCE morphologies at its two low energy states. The result is a very interesting set of motion figures with a perpetual character.

One advice for further improvement would be to explain in the main text better the fabrication procedure. Now I had to read this in the SI, but the fibre preparation is a very essential part for understanding of the mechanisms.

For the remaining, the paper is well written and provided with adequate references.

Few minor details:

The authors used the word 'twistless' a few times in their text without a definition what this means in the context of the materials or fibers they make. This might deserve some explanation.

The paper focuses on the mechanical effects. I would have appreciated some information on the degree of order, e.g. as measured by X-ray prior and during actuation. Even more interesting would be to know the gradient in order parameter over the fibre cross-section but I admit that would be difficult to quantify.

Dirk Broer

Reviewer #2 (Remarks to the Author):

In this manuscript, the authors present an optically-triggered fiber-type actuator, which is claimed to be phototunable, and self-oscillating. It is claimed that the present structure resembles that of helical geometries found in plants.

The device design follows three criteria, which in summary are: being able to show optically-triggered, straight-helical reversible transformation capable, capable of doing work.

The manuscript includes interesting results, but the flaws, open questions, and lack of significant/meaningful impact are much heavier. The following elaborates on this.

First, the flaws:

1. The authors are not providing enough quantitative details in their measurements. When you are dealing with optical signals, you need to describe very well your beam and sensors. What type of light was used?, wavelength?, what type of beam?, the sensor resolution?, beam profile?, how was the light intensity determined?, if a thermal camera was used, what is the assumed emissivity of the material? what emissivity was used as reference for calibration? Etc. Not being descriptive in this part, puts a huge question mark in all the result analysis and even the conclusions. Without detailed, careful, descriptive measurements, the results can all be considered "qualitative" or "speculative" at best, and "unsubstantiated/inconclusive observations" at worst.

2. The authors claim that "artificial oscillating systems capable of transducing simple constant input into complex motion that synchronizes different oscillations have not yet been reported". This is simply not true. There are many phenomena that prove this statement wrong, perhaps the most trivial being a simple pendulum (!), where gravity functions as a "simple constant input".

Next, the open questions:

1. The actuation process here is not described clearly. The manuscript mentions "photothermal excitation of the dopants" and "photo-thermal-mechanical transfer"; but an absorption spectrum for the light used is not shown. In fact, not even the absorption coefficient is given (!) --not to mention the wavelength used.

2. The authors claim that near infrared was used to actuate these devices. However, the light spot is clearly visible in the camera --as well as the structure and everything else that should be visible to the human eye. This begs a careful explanation. Although there are cameras that can see both wavelengths (visible and NIR), it is very intriguing to see both the objects and the beam so clear in the videos and images.

Finally, the significance of the work:

1. Essentially, the present work shows an optically-controlled mechanical oscillator. There is vast studies and prior art with much more detailed, conclusive, and deep scientific studies. In order for the present work to represent an advancement in the field, the studies need to be far more detailed and quantitative; which include descriptive experiments that demonstrate improvements over the current state-of-the-art.

Reviewer #3 (Remarks to the Author):

Hu et al. describe the photothermally-powered oscillation of a liquid crystal elastomer that morphs from a fiber to a spring. The fundamental mechanism of this oscillation is selective heating (irradiation) of the actuator, which in turn leads to repeated deformation and recovery cycles in response to a constant stimulus. At the highest level, similar mechanisms have been reported in other oscillating photoresponsive materials where the reversible motion of portions of the film from light to dark are critical. However, this work focuses on actuating fibers (as compared to films). Enabled by this change, the properties that are demonstrated are impressive. Specifically, three modes of oscillation are demonstrated and each of these modes of oscillation happens at relatively high frequency for thermal actuators (> 1 Hz and sometimes > 10Hz). The oscillation occurs while

lifting a weight that is ~20x heavier than the actuator. Finally, two demonstrations of potential applications are included. I expect this work to be impactful in the field of soft actuators. There are some concerns that should be addressed prior to publication. The most important of these concerns is that the methods are insufficiently detailed.

1. The authors state "previous oscillating systems fail to work under loading". This is an overstatement. For example, work by Broer and coworkers has shown oscillators that locomote when attached to a passive frame (i.e. a load). A more nuanced commentary on the literature is needed.

2. The authors note the importance of oscillators that perform mechanical work. Can the work performed by these oscillators be added to the manuscript?

3. The characterization methods are not sufficiently described in order for the work to be repeated. For example, the tensile testing rate is not given. The intensity of the concentrated sunlight is not given. Figure S11 says that: "The mass of the fiber and weight are 1.5 mg and 73 mg, respectively. The mass of the fiber and weight are 1.5 mg and 62 mg, respectively." There are many more details that should be added beyond what I have named.

4. A screw mold is used to make the actuator, but the characteristics of the mold are not given.

Response to referee 1

Comments 1:

I read the paper of Zhiming Hu on self-oscillating of spiral liquid crystal polymer fibers with great interest. The authors produced a spring-shaped actuator that is capable to provide continuous oscillation under a constant light source. The actuation figures could be adjusted as winding/unwinding and a continuous linear displacement. The authors based their system on by itself known liquid crystal oligomeric materials used by others for the fabrication of actuating systems (see the many liquid crystal elastomer (LCE) publications also mentioned by the authors. To make the actuators respond to light the authors blended the LCE oligomer with graphene to convert light absorption into heat, inducing order variations in the polymer. A special element in the system, which to my opinion deserves publication in Nat Comm, is the use of a screw-shaped mould to pre-structure the LCE fiber and the two-stage crosslinking basically freezing the LCE morphologies at its two low energy states. The result is a very interesting set of motion figures with a perpetual character.

Response to referee:

We highly appreciate your professional evaluation.

Comments 2:

One advice for further improvement would be to explain in the main text better the fabrication procedure. Now I had to read this in the SI, but the fibre preparation is a very essential part for understanding of the mechanisms. For the remaining, the paper is well written and provided with adequate references.

Response to referee:

Thank you for your constructive advice. We have added the content to illustrate the fabrication procedure in the main text. The added content is shown below:

“we start the fabrication using a precursor of helically coiled soft spring (Fig. 1d and 1e) that possesses a differential in length between its outer and inner circumference (Fig. 1f). This precursor is prepared through employing a screw mold (Supplementary Fig. 1) to shape a liquid crystal elastomer (LCE) oligomer into the helically coiled soft spring with a well-defined weak network formed by chemical crosslink reaction (see details in the methods). After initial curing, the shaped soft spring, in which the chemical crosslink reaction is not completed and still in progress over time, is removed from the screw mold and used as the precursor. Next, the fresh prepared precursor is axially stretched (Fig. 1g). Upon the stretching, the stress accumulated in the inner side is much greater than the outer side of the soft spring because of their difference in length. After the stretching operation, the stress gradient over the cross-sections of the straightened spring fiber is induced and fixed via the chemical crosslink reaction.”

Comments 4:

The authors used the word ‘twistless’ a few times in their text without a definition what this means in the context of the materials or fibers they make. This might deserve some explanation.

Response to referee:

Thank you for your constructive advice. Helical structures can be induced in fibers mainly through

two distinct mechanisms. One uses twisting. For an example, if we take a piece of elastic fiber, hold it between fingers, and twist its ends, the fiber will soon coil on itself and form a helical structure. This is arising from the writhing instability where a local change in twist eventually results in a global reconfiguration of the fiber. In this case, we have a twist-to-writhe conversion ^[1]. The word writhe refers to global deformation of a fibrous structure.

In the contrast, the other mechanism, widely employed by plants in Nature (e. g. Tendril), relies on the asymmetric stress/strain across the cross-section of a fiber, which leads to the intrinsic curvature and results in helical structure. The fabrications using this mechanism does not have any twisting operation, therefore the prepared helical fibers are often called twistless spring fibers.

To clearly illuminate this point, we have revised the related text. The revised text is shown below:

“Whereas plant tendrils, on the contrary, show a twistless mechanism that relies on the intrinsic curvature rather than external twisting.”

[1] McMillen, T. & Goriely, A. Tendril perversion in intrinsically curved rods. *J. Nonlinear Sci.* **12**, 241-281 (2002).

Comments 5:

The paper focuses on the mechanical effects. I would have appreciated some information on the degree of order, e.g. as measured by X-ray prior and during actuation. Even more interesting would be to know the gradient in order parameter over the fibre cross-section but I admit that would be difficult to quantify.

Response to referee:

Thank you for your constructive advice. We have measured the order of degree by X-ray prior and during actuation. As shown in **Fig. S1**, before light irradiation, the symmetrically distributed diffraction spots in the wide-angle area can be clearly observed, which suggests that the fiber prepared by stretching treatment has uniaxial orientation. When the fiber is irradiated and triggered, the diffraction spots in the wide-angle area become diffuse, indicating that the degree of order of the fiber is reducing. As the light intensity increases, the degree of order gradually decreases. And the degree of order is inversely proportional to the intensity of the incident light.

Fig. S1. Degrees of order in spring fiber actuator. **a, b**, Optical photographs and corresponding 2D XRD patterns of a piece of the fiber prior and during actuation. **c**, Evolution of the degree of order in the fiber along with increase of light intensity.

In addition, to measure the order parameter over the fibre cross-section, four areas at equal distance interval from the inner side to the outer side of the fiber were tested by 2D XRD (Fig. S2a and S2b). As shown in the Fig. S2b, 2D XRD pattern of the inner side of the fiber clearly shows symmetrical diffraction spots, which indicates high degree of order. As the X-ray spot moves from the inner side to the outer side, the diffraction spots become diffuse, suggesting that the degree of order in the fiber gradually decreases from the inner side to the outer side, and a gradient of order parameter exists over the fibre cross-section (Fig. S2c).

Fig. S2. Gradient in order parameter over the cross-section of the fibre. a. photograph showing four areas enclosed by the four colored circles at equal distance interval from the inner side to the outer side of the fiber, which were irradiated and tested in the 2D XRD measurement. **b.** 2D XRD patterns for the four areas on the fiber. **c.** Evolution of the degree of order over the fibre cross-section.

Response to referee 2

Comments 1:

In this manuscript, the authors present an optically-triggered fiber-type actuator, which is claimed to be phototunable, and self-oscillating. It is claimed that the present structure resembles that of helical geometries found in plants. The device design follows three criteria, which in summary are: being able to show optically-triggered, straight-helical reversible transformation capable, capable of doing work.

Response to referee:

Thank you for your comments.

Comments 2:

The manuscript includes interesting results, but the flaws, open questions, and lack of significant/meaningful impact are much heavier. The authors are not providing enough quantitative details in their measurements. When you are dealing with optical signals, you need to describe very well your beam and sensors. What type of light was used?, wavelength?, what type of beam?, the sensor resolution?, beam profile?, how was the light intensity determined?,

Response to referee:

We are pleased that the reviewer found the results of the manuscript interesting. We are sorry that we did not provide enough quantitative details. In the revised manuscript, we have added contents to quantitate our experiments. The added contents have been highlighted by yellow color in the revised supplementary manuscript.

The light used to actuate the fiber is 808-nm NIR laser, which is produced by two types of semiconductor laser diodes. One uses AlGaInP semiconductor (laser pointer, HW808AD1200-22FGD, Shenzhen Infrared Laser Technology Co., Ltd), the other employs AlGaAs semiconductor (PSU-H-LED, MDL-H-808-5W, Changchun New Industries Optoelectronics Technology Co., Ltd). We measured the profiles of these two lasers by a camera-based beam profiler (BeamOn-U3-VIS-NIR, Duma Optronics Ltd). Laser intensities were measured by a high precision laser power meter (TP100, Changchun New Industries Optoelectronics Technology Co., Ltd). We also carried out laser spectral analysis by a high-resolution spectrometer (Aurora4000, Changchun New Industries Optoelectronics Technology Co., Ltd.). The results of the measurements are shown below:

Fig. S3. Beam profiles of two 808-nm lasers generated by the semiconductor laser diodes of AlGaInP (a) and AlGaAs (b), respectively. The curves near the photos show the intensity distributions of the lasers.

Fig. S4. Laser spectral analysis of two 808-nm laser beams generated by the semiconductor laser diodes of the AlGaInP (a) and AlGaAs (b), respectively.

Three lights used to demonstrate the steering function exhibited in Fig.5 in the manuscript are 450-nm laser (blue), 532-nm laser (green), and 650-nm laser (red), which are gained by commercial laser pointers purchased from Beijing Huisite Technology Co., Ltd (Green and red laser pointers) and Shenzhen Xinfei Optoelectronics Technology Co., Ltd (Blue laser pointer). The beam profiles of the three lasers as well as their spectral analysis are shown below:

Fig. S5. Beam profiles of 450-nm laser (a), 532-nm laser (b), and 650-nm laser (c). The curves near the photos show the intensity distributions of the lasers.

Fig. S6. Laser spectral analysis of 450-nm laser (a), 532-nm laser (b), and 650-nm laser (c).

Comments 3:

If a thermal camera was used, what is the assumed emissivity of the material? what emissivity was used as reference for calibration? Etc. Not being descriptive in this part, puts a huge question mark in all the result analysis and even the conclusions. Without detailed, careful, descriptive measurements, the results can all be considered "qualitative" or "speculative" at best, and "unsubstantiated/inconclusive observations" at worst.

Response to referee:

Thank you for your valuable comments. In our experiment, a thermal camera (A665sc, FLIR) was used to monitor the temperature change of the oscillating fiber actuators. The emissivity set of this camera is 0.96. This camera of FLIR is calibrated to factory specifications before sent to users.

Calibration is performed under controlled conditions with a large number of blackbody reference sources. Blackbodies are physical bodies with very high emissivity, meaning they radiate and absorb almost all electromagnetic radiation. The blackbodies in a calibration lab are certified and traceable to international standards. The blackbody reference sources are arranged in a semi-circle and set to different known temperatures, and then the thermal camera (connected to a robotic arm) is pointed at each reference source one by one. The signal value at each temperature is captured by calibration software, and each pair of signal and temperature values are plotted along a curve, the equation of which is based on a physics model. This data is then loaded into the camera, calibrating it to ensure it meets accuracy specifications [2]. The thermal camera of FLIR must be periodically sent to the manufacturer for the calibration since a professional lab with the extensive requirements is needed.

According to the user manual of FLIR, before starting experiments, we can perform a simple calibration check to ensure that a thermal camera measures accurate temperature. Calibration checks are performed by measuring targets with known temperatures and comparing the known vs. the measured temperature. In this case, we can use boiling water and melting ice. Boiling water will have a temperature of about 100°C. Melting ice will have a temperature of about 0°C. Set camera's emissivity set to 0.96 and point it at these targets to take a measurement of the temperature to verify whether the camera works accurately. The result of our calibration checks is shown below (Fig. S7), which confirms that the measurement of our thermal camera is accurate.

Fig. S7. Calibration check performed by measuring boiling water (100 °C) and melting ice (0 °C) with known temperatures and comparing the known vs. the measured temperature.

[2] <https://www.flir.com/discover/professional-tools/how-do-you-calibrate-a-thermal-imaging-camera/>

Comments 4:

(2) *The authors claim that "artificial oscillating systems capable of transducing simple constant input into complex motion that synchronizes different oscillations have not yet been reported". This is simply not true. There are many phenomena that prove this statement wrong, perhaps the most trivial being a simple pendulum (!), where gravity functions as a "simple constant input".*

Response to referee:

Thank you for your valuable comments. We are sorry that we did not express it accurately. Here, we would like to emphasize that the manmade light-driven synchronization of diverse oscillations

in one oscillating system is rarely reported [3]. We have revised the related text to make it accurate. The revised contents are shown below:

“To our knowledge, artificial light-driven oscillating systems capable of transducing simple constant input into complex motion that synchronizes different oscillations have been seldom reported [3].”

[3] Vantomme, G. et al. Coupled liquid crystalline oscillators in Huygens' synchrony. *Nat. Mater.*, <https://doi.org/10.1038/s41563-021-00931-6> (2021).

Comments 4:

The actuation process here is not described clearly. The manuscript mentions "photothermal excitation of the dopants" and "photo-thermal-mechanical transfer"; but an absorption spectrum for the light used is not shown. In fact, not even the absorption coefficient is given (!) --not to mention the wavelength used.

Response to referee:

Thank you for your constructive advice. More details have been added in the manuscript. 808-nm NIR light was used to actuate the LCE fiber. The absorption spectra of the LCE film with (2 wt%) and without graphene is shown in Fig. S8. From these spectra, we can obtain the absorbance (A) of the LCE film with and without graphene at the wavelength of 808 nm.

One of the important parameters determining the optical properties of a material is the optical absorption coefficient (α), which demonstrates the ability of a material to absorb the light of a given wavelength. The change of α according to A is given by [4],

$$\alpha = 2.303 \frac{A}{d} \quad (1)$$

Where A is the absorbance, and d is the thickness of the sample. The wavelength dependence of the α value obtained by using Eq. 1 are given in Fig. S8c. The optical absorption coefficient (α) of the LCE film with (2 wt%) and without graphene at the wavelength of 808 nm is exhibited in Table S1.

Fig. S8. Optical absorption spectra. a, b, Absorption spectra of LCE film with (2 wt%) and without graphene. **c, d,** Evolution of absorption coefficient of LCE film with (2 wt%) and without graphene along with increase of wavelength.

Table S1. Absorption coefficient of the LCE film with and without graphene

Graphene mass fraction	Film thickness ^a (μm)	Coefficient of Absorption (cm ⁻¹)
0%	4.44 ±0.11	93.4
2%	3.28 ±0.15	576.5

^a Average value of film thickness of three different position on the film.

[4] Mergen, Ö. B., Arda, E., Kara, S. & Pekcan, Ö. Effects of GNP addition on optical properties and band gap energies of PMMA films. *Polym. Composite.*, 2019, 40, 1862-1869.

Comments 5:

The authors claim that near infrared was used to actuate these devices. However, the light spot is clearly visible in the camera --as well as the structure and everything else that should be visible to the human eye. This begs a careful explanation. Although there are cameras that can see both wavelengths (visible and NIR), it is very intriguing to see both the objects and the beam so clear in the videos and images.

Response to referee:

Thank you for your important comment. The light used to actuate the fiber is 808-nm NIR light, which is produced by semiconductor laser diodes. The beam of 808-nm NIR light is invisible to human eyes, but its light spot can be observed when the relatively high intensity of the light is used, which can be verified by previous studies in which the spots of 808-nm NIR light are visible ^[5, 6].

The lasers employed to demonstrate the beam steering function shown in Fig. 5 in the manuscript are 450-nm (blue), 532-nm (green), and 650-nm (red) lasers, which are generated by commercial laser pointers. Indeed, in most daily environments, we cannot see these laser beams. To make them visible in our experiments, we made an artificial dark environment full of suspended particles. The laser beams were scattered by the suspended particles, the scattered light makes us see the beams. In addition, the dark environment can effectively enhance our visual sensitivity to the scattering light. As a result, we can clearly see these laser beams through their scattering lights in the specific environment.

[5] Wang, E., Desai, M. S. & Lee, S. W. Light-controlled graphene-elastin composite hydrogel actuators. *Nano Lett.* **13**, 2826-2830 (2013).

[6] Zuo, B., Wang, M., Lin, B. P. & Yang, H. Visible and infrared three-wavelength modulated multi-directional actuators. *Nat. Commun.* **10**, 4539 (2019).

Comments 6:

Essentially, the present work shows an optically-controlled mechanical oscillator. There is vast studies and prior art with much more detailed, conclusive, and deep scientific studies. In order for the present work to represent an advancement in the field, the studies need to be far more detailed and quantitative; which include descriptive experiments that demonstrate improvements over the current state-of-the-art.

Response to referee:

We appreciate the comment of the reviewer. We have added contents to detail and quantitate the studies. The added contents have been highlighted by yellow color in the revised manuscript.

Our work presents a novel phototunable self-oscillating system. *Active tuning with a capability of local manipulation to achieve diverse self-oscillations in a single actuator is the unique feature of our system.* We kindly request the reviewer to reconsider the unique features of our work in a broader context as outlined below:

- (1) **A novel phototunable self-oscillating system that possesses a broad range of oscillation modes, and controllable evolution between diverse modes.** In the previously reported oscillators (*Adv. Mater.*, 2020, **32**, 1906319; *Adv. Mater.*, 2017, **29**, 1606712; *Soft Matter*, 2010, **6**, 779), bending is largely shown as main degree of freedom (DOF), these oscillators only exhibit single mode of self-oscillation. *It has proven extremely challenging to achieve controllability of various oscillations in a single actuator* [*Nat. Commun.*, 2019, **10**, 5057]. **Our system enables not only controllable generation of three basic self-oscillations but also production of diverse complex oscillatory motions in one actuator.**
- (2) **A twistless approach to prepare a self-winding fiber actuator with high degree of freedom.** Widely used approaches to introduce the structure of helical coils is through twisting insertion of highly stretched orientated polymer fiber. Whereas in our work, we employ a twistless approach that make use of the intrinsic curvature generated from an asymmetrical distribution of strain/stress over the cross-sections of the fiber to induce helix structure, which enable generation of diverse deformation behaviors with high DOF, such as bending, twisting, coiling, and winding.
- (3) **High Robustness of autonomous self-oscillation of our system.** Our system can work continuously over million cycles without obvious fatigue, which ensures long-term reliability. This is critically important when a self-oscillating system would like to be integrated into consumer-based products.

Response to referee 3

Comments 1:

Hu et al. describe the photothermally-powered oscillation of a liquid crystal elastomer that morphs from a fiber to a spring. The fundamental mechanism of this oscillation is selective heating (irradiation) of the actuator, which in turn leads to repeated deformation and recovery cycles in response to a constant stimulus. At the highest level, similar mechanisms have been reported in other oscillating photoresponsive materials where the reversible motion of portions of the film from light to dark are critical. However, this work focuses on actuating fibers (as compared to films). Enabled by this change, the properties that are demonstrated are impressive. Specifically, three modes of oscillation are demonstrated and each of these modes of oscillation happens at relatively high frequency for thermal actuators (> 1 Hz and sometimes > 10 Hz). The oscillation occurs while lifting a weight that is ~ 20 x heavier than the actuator. Finally, two demonstrations of potential applications are included. I expect this work to be impactful in the field of soft actuators.

Response to referee:

We highly appreciate your professional comments.

Comments 2:

There are some concerns that should be addressed prior to publication. The most important of these concerns is that the methods are insufficiently detailed.

Thank you for your constructive advice. We have added more details in the methods, which are highlighted by yellow color in the manuscript.

Comments 3:

The authors state "previous oscillating systems fail to work under loading". This is an overstatement. For example, work by Broer and coworkers has shown oscillators that locomote when attached to a passive frame (i.e. a load). A more nuanced commentary on the literature is needed.

Response to referee:

Thank you for your constructive advice. We have revised the statement to make it accurate. The revised text is shown below:

“Most previous oscillating systems fail to work under loading.”

Comments 3:

The authors note the importance of oscillators that perform mechanical work. Can the work performed by these oscillators be added to the manuscript?

Response to referee:

Thank you for your valuable advice. To make an oscillator perform mechanical work, our design strategy is to integrate the external load as a part of our oscillating system, like the classical block-spring oscillator. The external load connects and works together with the “soft spring” (SWFA) to generate oscillating motions. As shown in Supplementary Video S1, the fiber actuator can load a magnetic bar, and perform mechanical work to lift the bar up-and-down in a metal coil. Through doing this mechanical work, the oscillating system can effectively transduce light energy into electricity.

Comments 4:

The characterization methods are not sufficiently described in order for the work to be repeated. For example, the tensile testing rate is not given. The intensity of the concentrated sunlight is not given. Figure S11 says that: "The mass of the fiber and weight are 1.5 mg and 73 mg, respectively. The mass of the fiber and weight are 1.5 mg and 62 mg, respectively." There are many more details that should be added beyond what I have named.

Response to referee:

Thank you for your valuable advice. We have added experimental details in the manuscript to clarify our characterization methods. In our experiments, the tensile testing rate is 50 mm/min. The intensity of the concentrated sunlight is $\sim 3.5 \text{ W cm}^{-2}$, which was measured by a solar power meter (LP-3A, Merry Change). The description of Figure S11 has been revised, which is shown below:

“The mass of the fiber and the load is 1.5 mg and 73 mg, respectively.”

Comments 5:

A screw mold is used to make the actuator, but the characteristics of the mold are not given.

Response to referee:

Thank you for your valuable advice. The characteristics of the screw mold have been added in the Supporting Information of the manuscript. The added content is shown below:

Fig. S9. The characteristics of the screw mold. a, Optical photographs showing the screw mold. **b,** Schematics showing the characteristic features of the screw mold.

REVIEWERS' COMMENTS

Reviewer #1 (Remarks to the Author):

I re-evaluated the manuscript of Hu et al and compared with the earlier version. They expanded especially on their supporting information and in their main text in the 'Methods' section (as requested by all reviewers). As indicated in my earlier report, I like the concept although photo-thermal actuation of two-stage crosslinked liquid crystal oligomers is not new and related to this one might argue whether the paper is innovative enough for nat. Comm. Nevertheless, the concept of making a spiral configuration, capable to undergo a variety of deformation figures is nice and appreciated.

The authors addressed the points I raised in my earlier report and with this respect I am satisfied.

Reviewer #2 (Remarks to the Author):

The authors have done a genuine effort in correcting the manuscript and have addressed all the comments in the previous review. I would argue that there is plenty of information in the response to authors letter that should also be included in the supplementary material of the manuscript (but it was not highlighted in yellow in the response letter).

Reviewer #3 (Remarks to the Author):

The authors have significantly improved the manuscript to address the concerns of each reviewer. I recommend that the article be published in its current form.

Response to referee 1

Comments 1:

I re-evaluated the manuscript of Hu et al and compared with the earlier version. They expanded especially on their supporting information and in their main text in the 'Methods' section (as requested by all reviewers). As indicated in my earlier report, I like the concept although photo-thermal actuation of two-stage crosslinked liquid crystal oligomers is not new and related to this one might argue whether the paper is innovative enough for nat. Comm. Nevertheless, the concept of making a spiral configuration, capable to undergo a variety of deformation figures is nice and appreciated. The authors addressed the points I raised in my earlier report and with this respect I am satisfied.

Response to referee:

We highly appreciate your professional evaluation.

Response to referee 2

Comments 1:

The authors have done a genuine effort in correcting the manuscript and have addressed all the comments in the previous review. I would argue that there is plenty of information in the response to authors letter that should also be included in the supplementary material of the manuscript (but it was not highlighted in yellow in the response letter).

Response to referee:

Thank you very much for your constructive advice. We have added the information as the supplementary note in the revised supplementary manuscript.

Response to referee 3

Comments 1:

The authors have significantly improved the manuscript to address the concerns of each reviewer. I recommend that the article be published in its current form.

Response to referee:

Thank you very much for your comment.